# The prevalence, severity and chronicity of abuse towards older men: Insights from a multinational European survey

Maria Gabriella Melchiorre[1], Mirko Di Rosa[2]*, Gloria Macassa[3,4], Bahareh Eslami[4], Francisco Torres-Gonzales[5], Mindaugas Stankunas[6,7], Jutta Lindert[8,9], Elisabeth Ioannidi-Kapolou[10], Henrique Barros[11], Giovanni Lamura[1], Joaquim J. F. Soares[4]

1 Centre for Socio-Economic Research on Ageing, National Institute of Health and Science on Aging, IRCCS INRCA, Ancona, Italy, 2 Laboratory of Geriatric Pharmacoepidemiology, National Institute of Health and Science on Aging, IRCCS INRCA, Ancona, Italy, 3 Department of Occupational and Public Health Sciences, University of Gävle, Gävle, Sweden, 4 Division of Public Health Sciences, Department of Health Sciences, Mid Sweden University, Mittuniversitetet, Sundsvall, Sweden, 5 Department of Psychiatry, Faculty of Medicine, University of Granada, Granada, Spain, 6 Department of Health Management, Lithuanian University of Health Sciences, Kaunas, Lithuania, 7 Health Service Management Department, School of Medicine, University of Griffith, Gold Coast, Queensland, Australia, 8 Department of Public Health, University of Emden, Emden, Germany, 9 Women's Studies Research Center, Brandeis University, Waltham, MA, United States of America, 10 Department of Sociology, National School of Public Health, Athens, Greece, 11 EPIUnit, Instituto de Saúde Pública da Universidade do Porto, Porto, Portugal

* m.dirosa@inrca.it

**Data Availability Statement:** All relevant data are available from Dryad Dataset, https://doi.org/10.5061/dryad.f7m0cfxtt.

## Abstract

### Background

Elder abuse is a growing public health question among policy makers and practitioners in many countries. Research findings usually indicate women as victims, whereas male elder abuse still remains under-detected and under-reported. We aimed to investigate the prevalence, severity and chronicity of abuse (psychological, physical, physical injury, sexual, and financial) against older men, and to scrutinize factors (e.g. demographics) associated with high chronicity of any abuse.

### Methods

Randomly selected older men (n = 1908) aged 60–84 years from seven European cities (Ancona, Athens, Granada, Kaunas, Stuttgart, Porto, Stockholm) were interviewed in 2009 via a cross-sectional study concerning abuse exposure during the past 12 months.

### Results

Findings suggested that prevalence of abuse towards older men varied between 0.3% (sexual) and 20.3% (psychological), with severe acts between 0.2% (sexual) and 8.2% (psychological). On the whole, higher chronicity values were for injury, followed by psychological, financial, physical, and sexual abuse. Being from Sweden, experiencing anxiety and having a spouse/cohabitant/woman as perpetrator were associated with a greater "risk" for high

**Funding:** The ABUEL Project, "Elder Abuse: A multinational prevalence survey." was supported by the European Commission, through the Executive Agency for Health and Consumers (EAHC, currently CHAFEA, Consumers, Health, Agriculture and Food Executive Agency, https://ec.europa.eu/chafea/index_en.htm), Public Health Programme 2008–2010 (Grant Agreement n. 2007123). The Project was awarded to JFS, FTG, MS, JL, EIK, HB, and GL. The funders had no role in study design, data collection and analysis, decision to publish, or preparation of the manuscript.

**Competing interests:** The authors have declared that no competing interests exist.

chronicity of any abuse. For men, severity and chronicity of abuse were in some cases relatively high.

## Conclusions

Abuse towards older men, in the light of severe and repeated acts occurring, should be a source of concern for family, caring staff, social work practice and policy makers, in order to develop together adequate prevention and treatment strategies.

## Introduction

Elder abuse is a crucial public health concern and it has been associated with several negative health outcomes such as injury, poor mental health (e.g. depression), low social support and decreased quality of life [1–6]. According to a systematic review, including primary research on general population, the overall prevalence of elder abuse varies between 3.2–27.5%, probably reflecting variation in assessing abuse rates across cultures, and this is due to various factors (e.g. socio-demographics characteristics) [7]. Further studies, among the general population, found that elder abuse rates vary between 0.2–27.5% [8–10]. According to a more recent systematic review and meta-analysis [11], including 52 studies in 28 countries, the pooled prevalence rate, for the overall elder abuse in community settings, is 15.7%, especially psychological (11.6%) and financial (6.8%), followed by neglect (4.2%), physical (2.6%), and sexual (0.9%) abuse.

Several studies regard Intimate Partner Violence (IPV) [12–15], but the specific issue of sex differences in all types of elder abuse has not generated much attention. In this respect some international studies suggested that older women are more likely to be abused than older men [4, 9, 10, 16, 17], whereas, conversely, others authors found that men are, in some cases, more likely to be abused than women [18–20]. According to further studies, no differences between genders emerged [8, 21–24]. Overall thus, some data indicates that also older men are victims of abuse, even though the focus is often on women as victims [25, 26]. Concerning the perpetrator's sex, women tend usually to be portrayed as loving and caring persons rather than violent, and men as the opposite. A study from the United States of America (USA) [27] showed that 52.5% of the episodes of abuse were perpetrated by men and 47.5% by women. However, another study regarding older African Americans [18] observed that women, compared to men, were more likely to mistreat older relatives (75% vs. 67%), and in a review of 200 studies, concerning IPV in all ages [15], a gender balance in perpetration was found.

Abuse against older men is a reality, but it remains little reported, probably because men feel shame and mortification of being abused, and also they fear further abuse. Moreover, they don't want to admit such an experience and to ask for help [28–33]. It has also been argued that, the ambiguous results concerning sex differences in elder abuse, may be due to methodological inconsistencies, e.g. operational definition of abuse and/or failure to control for important factors such as abuse severity [34–38].

Interestingly, there are on the whole little data on the occurrence of different types of abuse (e.g. psychological) by chronicity (frequency of acts) and severity (minor and severe acts), as well as, in particular, on the factors influencing chronicity among older men, despite the fact that it has been stressed the importance of such data to better understanding of abuse [e.g. 39]. In this respect a study, however regarding only women, indicated that almost 6% of those aged 60 years and over, in five European countries, experienced multiple forms of abuse "very

often" [25] and another found that the frequency of abuse (chronicity) was high in Sweden, compared to six other European countries [6]. Furthermore, it has been suggested that, when older persons are subjected to frequent and multiple abusive acts, they are more likely to experience hospitalization [40], all-cause mortality [2] and poor health [41].

Considering the above-mentioned findings and paucity of data on abuse, regarding old male victims, this study aimed to: (1) describe the prevalence, chronicity and severity of psychological, physical, injury, sexual, financial and any abuse type experienced by older men in seven European countries, during the past year; (2) examine factors (e.g. socio-demographics) associated with total high chronicity (minor/severe acts) of any abuse type among older men.

## Materials and methods

### Study design and ethics statement

The present paper was based on data from the main study "ABUEL" (ELder ABUse: A multi-national prevalence survey). It was a multinational and cross-sectional prevalence survey on elder abuse, which during January-July 2009 was conducted by face-to-face and self-administered interviews to older people, in seven urban European cities (Ancona, in Italy; Athens, in Greece; Granada, in Spain; Kaunas, in Lithuania; Stuttgart, in Germany; Porto, in Portugal, and Stockholm, in Sweden) [5, 42]. For the sake of simplicity, reported/discussed findings refer to the related countries instead of cities where the studies took place, by assuming these cities exploratory/pilot examples in this respect.

Respondents were first contacted by telephone/letter, and then an appointment was set. They were interviewed usually in their homes, and ensuring that they were alone. The interviews focused on abuse exposure (and further aspects, e.g. health, lifestyle, and relationships) during the past 12 months, and were conducted by trained interviewers, according to a detailed user's guide. In two cities (i.e. Stockholm, Stuttgart), respondents who did not want to have a face-to-face interview received a questionnaire for self-response. All survey materials (e.g. questionnaire, information letters), including measures/tests if not already available/validated, were translated into the native languages, and culturally adapted. In this respect, the translation process of the questionnaires, from English into the native language in each country, was provided by following detailed guidelines, including translation and back translation (for linguistic and cross-cultural validation of the standardized assessment tool), support from a review committee to solve criticisms, and pilot testing of a couple of interviews in each country. During the pilot phase, each partner documented country discrepancies due to particular issues, such as services used, education, occupational status, and sources of income. All the possible question changes, with respect to the final version of the questionnaire in English, have been recorded in the translation process documentation, and some cultural adaptations were necessary in order to substitute any "ambiguous" term (in English) with others which met better each country/cultural context, and related usual experiences or activities, anyway maintaining the whole general meaning of the original labels. Items might indeed be equivalent in semantic meaning but not conceptually. Moreover, each interviewer had a guide where the meaning of some terms in his/her own country was fully explained/described (e.g. education categories, professional groups, main source of financial support, and different available services).

The respondents were fully informed about the aim of the study, and written informed consent forms were obtained prior to data collection from all participants. Confidentiality, anonymity and the participant's rights were emphasized. The respondents could stop the data gathering at any point in time. The study was approved by national/university or regional ethics committees in each participating country, except for Greece where the fieldwork was

carried out by the QED Company, which is member of ESOMAR (European Society for Opinion and Marketing Research), and provides global guidelines for ethics [5].

The full names of the other six ethics committees/institutional review boards [5] were the following: Regional etisk kommittee vid Karolinska Institutet (Karolinska Institute, Regional Ethics Committee), in Sweden; Ethikkommission des Landes Baden-Wuerttemberg (Ethics Committee of the State of Baden-Wuerttemberg), in Germany; Comitato di Bioetica INRCA, Istituto Nazionale di Riposo e Cura per Anziani, Ancona (National Institute of Health and Science on Ageing, Bioethics Advisory Committee), in Italy; Kauno regioninio biomedicininiu tyrimu etikos komitetas (Kaunas Regional Research Ethics Committee), in Lithuania; Comité de Ética do Hospital de João, Porto (Ethics Committee of the John Hospital, Porto), in Portugal; Comité de Etica en Investigación de la Universidad de Granada (Research Ethics Committee, University of Granada), in Spain.

## Participants

Participants were randomly selected from the general population (census/registry-based). They were: community-dwelling older women and men aged 60–84 years; living in own/ rented housing or homes for elderly people; and citizens or documented (self-report) migrants. Moreover, individuals not suffering from cognitive or sensory impairments assessed by the Mini-Cog [43], and able to read/write or express themselves in their native languages, were included in the survey.

Based on a mean abuse prevalence of 13%, derived from a previous review [7] with a precision of 2.6%, a total sample size of 633 individuals in each city was required, but a maximum of 656 individuals was allowed, in view of the infinite population assumption. The size of the sample was adjusted to each city, according to respective total population of women/men aged 60–84 years. The respondents were thus chosen by random stratification by sex/age. The final sample comprised 4467 older persons (57.3% women), with a mean response rate across countries of 45.2%. For the present study, the focus was only on men (n = 1908).

## Measures

Participants completed a standardized questionnaire with a set of validated instruments.

Abuse was assessed with 52 items, based on the Conflict Tactic Scales-2 (CTS2) [44] and the UK survey of elder abuse/neglect [45]. Psychological abuse comprised 11 items, of which 6 were severe acts (e.g. threatened with being hit or having something thrown at) and 5 minor (e.g. shouted or yelled at). Physical abuse had 17 items, of which 10 were severe acts (e.g. burned or scalded) and 7 minor (e.g. being grabbed). Injury had 7 items, of which 4 were severe acts (e.g. passed out from being hit on the head) and 3 minor (e.g. had a sprain, bruise or small cut from being hit). Sexual abuse had 8 items, of which 5 were severe acts (e.g. had sexual intercourse with you against your will) and 3 minor (e.g. tried to touch you in a sexual way against your will). Finally, financial abuse comprised 9 items, of which 5 were severe acts (e.g. made you give her/him/them your money, possessions or property against your will) and 4 minor (e.g. tried to make you give money, possessions or property). If there were more types of acts of abuse occurred in one time (e.g. both minor and severe), the prevalence was assessed by counting "one" for each single event.

Abuse chronicity (frequency of acts) was expressed as follows: occurred once (1), twice (2), 3–5 (midpoint 4), 6–10 (midpoint 8), 11–20 (midpoint 15) or >20 (midpoint 25) times, during the past year. Respondents were considered as abuse cases (yes = 1) when they referred the experience of at least one single episode/event of abuse, and the related frequency, during the past 12 months. The present study concentrated on prevalence, overall chronicity and high

chronicity by severity (minor, severe), and total of each abuse form (e.g. psychological) and any abuse (all types). Based on the abuse frequency (1, 2, 4, 8, 15, 25 times) of the total abused male population, medians of chronicity by severity (minor, severe, total), of each abuse form and any abuse, were also calculated. Thereafter, chronicity was dichotomized in low (abuse frequency under/on median) and high (abuse frequency above the median).

Depression and anxiety were assessed with The Hospital Anxiety and Depression Scale (HADS) [46]. It consists of 14 items (graded 0–3), of which 7 involve depression (e.g. lost interest in appearance), and 7 regard anxiety (e.g. sudden feelings of panic). Score ranges are from 0–21 for each scale. High scores correspond to high depression/anxiety levels. Scores 0–7 correspond to no cases, 8–10 to possible cases, and 11–21 to probable cases. For this study, the focus was on the total scores.

Somatic symptoms were assessed with the short 24-item version of the Giessen Complaint List (GBB) [47]. It consists of 24 items (graded 0–4, from not affected to very much affected), which are grouped in 4 domains of physical complaints (6 items each): exhaustion (e.g. tiredness), gastrointestinal (e.g. nausea), musculoskeletal (e.g. pains in joints or limbs), and heart distress (e.g. heavy, rapid or irregular heart-throbbing). Total somatic symptoms score range is from 0–96. The higher the scores, the more one is affected. Based on the median, total scores were dichotomised into low (under/on median) and high (above median) somatic symptom levels. For this study, the focus was on the total scores (not domains).

Social support was assessed with The Multidimensional Scale of Perceived Social Support (MSPSS) [48]. It consists of 12 items (graded 1–7), which are grouped into 3 sub-categories (4 items each): support from family, significant others, and friends. The total score range is from 12–84. High scores correspond to high perceived social support. Based on the median, total scores were dichotomised in low (under/on median) and high (above median) levels of social support. This study focused on the total scores.

Lifestyle variables included alcohol use and smoking. Alcohol use was assessed with items derived from The Alcohol Use Disorders Identification Test (AUDIT) [49]. For the present study, the participants were asked if they presently used alcohol (*do you drink alcohol*? yes/no). Regarding smoking (cigarettes) participants were asked if they presently smoked (*do you smoke*? yes/no?)

Demographics/socio-economics were assessed, and focused on the following variables: country; age (5 years groups; 60–64, 65–69, 70–74, 75–79, 80–84); marital status (i.e. married/cohabiting, alone, widowed, divorced/separated); ethnic background; educational level (i.e. low = informal/primary/similar; middle = high school/equivalent; high = university/similar); still on work; current or past (e.g. for retired people) main profession (blue-collar, low white-collar, middle/high white collar.); main source of financial support (i.e. pension, work, other income); with whom the participants lived (e.g. alone, spouse, cohabitant, other); and number of people at home, including the respondent (nuclear *vs.* extended family). Financial strain (concerns with how to make ends meet) was assessed with one item (possible answers were no/sometimes/often/always). Participant were considered to experience "financial strain" if they selected any response other than no. Four items (e.g. birth place) assessed whether the participants were migrants or indigenous inhabitants. The demographic and socio-economic variables were adapted for each country but were similar in content.

For the present study, perpetrator variables were assessed in form of sex (women/men), and relation to the victim (spouse/cohabitant *vs.* others, e.g. children/grandchildren, other relatives, friends, neighbors, caring staff), in a yes/no format. In this respect, it is to be highlighted that we analyzed various types of elder abuse (i.e. psychological, financial, physical, sexual, and injuries), but not specifically IPV, that is "any act of physical, sexual, psychological or economic violence that occurs between former or current spouses or partners, whether or not the

perpetrator shares or has shared the same residence with the victim" [50]. In general, we however considered as elder abuse perpetrators only spouse/cohabitant vs. others, because in the whole study/sample the spouses/partners were the most common perpetrators of psychological abuse (that was also the most perpetrated form of abuse), physical abuse, and injuries (respectively, about 35%, 34% and 45%), as shown in a previous publication from the same ABUEL study [51]. In our study we also found that married/cohabitant were 80%, and in 77% of cases persons at home were 1–2 (below in Table 1).

Further details regarding the whole Materials and Methods section (study design, participants, and measures) have been published by authors elsewhere [5, 33, 42].

**Table 1. Demographic/socio-economics, lifestyle and health characteristics of older men.**

| Variables | n (1908) | % |
|---|---|---|
| *Country* | | |
| Germany (Stuttgart) | 305 | 16.0 |
| Greece (Athens) | 287 | 15.0 |
| Italy (Ancona) | 270 | 14.0 |
| Lithuania (Kaunas) | 225 | 11.8 |
| Portugal (Porto) | 256 | 13.4 |
| Spain (Granada) | 272 | 14.3 |
| Sweden (Stockholm) | 293 | 15.4 |
| *Age* | | |
| 60–64 | 506 | 26.5 |
| 65–69 | 486 | 25.5 |
| 70–74 | 405 | 21.2 |
| 75–79 | 306 | 16.0 |
| 80–84 | 205 | 10.7 |
| *Foreign background* | | |
| No | 1809 | 94.8 |
| Yes | 94 | 4.9 |
| *Marital status* | | |
| Married/cohabitant | 1537 | 80.6 |
| Alone/single | 85 | 4.5 |
| Divorced/separated | 100 | 5.2 |
| Widower | 186 | 9.7 |
| *Education* | | |
| Low[a] | 613 | 32.1 |
| Middle[b] | 788 | 41.3 |
| High[c] | 470 | 24.6 |
| *Profession* | | |
| Blue-collar | 707 | 37.1 |
| Low white-collar | 465 | 24.4 |
| Middle/high white-collar | 708 | 37.1 |
| *Still on work* | | |
| No | 1503 | 78.8 |
| Yes | 404 | 21.2 |
| *Financial support* | | |
| Pension | 1469 | 77.1 |
| Working | 301 | 15.8 |
| Other income | 136 | 7.1 |

(*Continued*)

**Table 1.** (Continued)

| Variables | n (1908) | % |
|---|---|---|
| *Financial strain* | | |
| No | 798 | 41.9 |
| Yes | 1106 | 58.1 |
| *Lives with* | | |
| Spouse/cohabitant | 1160 | 60.8 |
| Spouse/cohabitant/other[d] | 395 | 20.7 |
| Alone | 248 | 13.0 |
| Other[e] | 97 | 5.1 |
| *Persons at home* | | |
| 1 to 2 persons | 1465 | 77.1 |
| More than 3 persons | 437 | 22.9 |
| *Alcohol use* | | |
| No | 441 | 23.1 |
| Yes | 1466 | 76.8 |
| *Smoking* | | |
| No | 1592 | 83.4 |
| Yes | 313 | 16.4 |
| *Somatic symptoms*[f,h] | | |
| High | 1156 | 60.6 |
| Low | 752 | 39.4 |
| *Anxiety symptoms* (Mean/SD) [i,m] | 3.99 | 3.55 |
| *Depression symptoms* (Mean/SD) [i,m] | 4.62 | 3.76 |
| *Social support*[g,l] | | |
| High | 1005 | 53.9 |
| Low | 860 | 46.1 |

[a] = Primary school/similar

[b] = Gymnasium/similar

[c] = University/similar

[d] = e.g. daughter/son, sister/brother, grandchildren

[e] = e.g. daughter/son, sister/brother, grandchildren

[f] = Scores higher than 35 in somatic symptoms are categorized as high

[g] = Scores higher than 70 in social support are categorized as high

[h] = GBB: Giessen Complaint List (range 0–85)

[i] = HADS: Hospital Anxiety and Depression Scale (range 0–21 for both dimensions)

[l] = MSPSS: Multidimensional Scale of Perceived Social Support (range 12–84)

[m] = Missing values for continuous variables were as: Depression: 24, Anxiety 20; SD = Standard Deviation.

## Reliability and validity of exposure variables

Internal consistency of exposure variables, as measure of reliability, was evaluated using the Cronbach's Alpha. Regarding abuse (CTS2 and items from UK survey), it was: for psychological 0.85, for physical 0.80, for sexual 0.76, for financial 0.64, and for injuries 0.70 [33]. Moreover, Ordinal Alpha, that is a more appropriate measure of internal consistency for scales with five or fewer options, was: for somatic symptoms (GBB) 0.95, for anxiety (HADS) 0.87, and for depression (HADS) 0.86. Finally, Cronbach's Alpha was 0.92 for total social support (MSPSS).

Validity, considered as estimate of the contribution of each individual item to the scale, was assessed using Pearson's correlation (between each item questionnaire scores and total scale

score). Pearson's correlation coefficients for abuse were: for psychological, between 0.27–0.72; for physical, between 0.13–0.71; for sexual, between 0.24–0.77; for financial, between 0.17–0.64; and for injuries, between 0.31–0.80. Moreover, Pearson's coefficients were: for somatic symptoms, between 0.40–0.74; for anxiety, between 0.64–0.75; for depression, between 0.56–0.77; for social support, between 0.68–0.77. Regarding low values for elder abuse validity, some factors could have exerted a negative role. For instance the following: our measure included CTS2 (a scale) with additional items from UK survey; and definitions/meaning of abuse may differ across cultures/countries [52, 53].

## Data analyses

Prevalence and chronicity of minor, severe, and total abuse types, experienced by older men, were calculated. The data on continuous variables were presented by means and Standard Deviation (SD), and categorical variables by absolute frequencies and percentages. Furthermore, a multiple block-wise logistic regression analysis was conducted for total high chronicity (minor vs. severe acts) of any abuse. In the block-wise logistic regression, variables were inserted into the regression equation block by block, and the contribution of every block, in explaining the dependent variable, was expressed as Nagelkerke $R^2$ changes. Nagelkerke $R^2$ is an approximation to descriptive goodness-of-fit statistics, to measure the fit of the proposed logistic model (to quantify the strength of connection between variables) [54].

With regard to the multiple block-wise logistic regression analysis, the dependent variable was total high chronicity of any abuse (psychological, physical, financial, sexual, and injury). The independent variables were selected based on previous analyses, that differentiated abused/non-abused respondents [e.g. 5, 6, 55], and were categorized in the following four main blocks:

1. Victims' demographic/socio-economic block, including country, age, foreign background, marital status, education, profession, still on work, financial support, financial strain, with whom the participants lived, and number of people at home;

2. Victims' life-style block, including alcohol use and smoking tobacco in yes/no format;

3. Victims' health block, including somatic symptoms (low/high), and possible/probable cases of depression/anxiety;

4. Victims' social support block, comprising social support (low/high);

Additionally, the perpetrators' block was included, regarding relationship with the victim (i.e. spouses/cohabitants, others), and sex of the perpetrator (women/men).

Associations between variables were expressed as Odds Ratio, 95% CI, and Nagelkerke $R^2$. Finally, it has been also scrutinized if the model fitted the data, with the Hosmer-Lemeshow test for logistic regression, which showed no evidence of poor fit. Missing values were excluded from the multivariable analyses (189 subjects, of which 25 in Sweden and 29 in Germany from self-administered questionnaires). The criterion for considering valid tests/scales was 100% of responses regarding MSPSS and HADS. The GBB missing items were directly recoded into "not at all" category (possible answer to the question: "*How much does each complaint discomforts you?*"). The analyses were carried out using the SPSS statistical package 22. The significant level was set at p < .05.

## Results

### Demographic/socio-economics, lifestyle and health characteristics

Table 1 depicts the demographics and socio-economic characteristics of old male participants.

The responses put mainly in evidence a greater prevalence of individuals aged 60–74 years, and also that only 5% had a foreign background, around 80% were married/cohabitant, and 41% had a middle educational level. With regard to the occupation, 37% of the sample were blue-collar and similarly 37% were middle/high white collar. Moreover, only 21% were still on work, 77% had a work pension as main financial support, and 58% reported financial strain. Regarding household, participants were mainly living with a spouse/cohabitant (61%), and persons in the home (including the respondent) were 1–2 in 77% of cases. Finally, older men of the sample were very often alcohol drinkers (77%) and "light" smokers (16%), with high somatic symptoms in 61% of cases, low mean values of anxiety and depression, and high social support in 54% of cases.

## Prevalence and severity of abuse

As shown in Table 2, the highest prevalence rates of total abuse pertained to psychological abuse (20.3%), followed by financial (4.2%) and physical abuse (2.8%), with injury (0.4%) and sexual abuse (0.3) as the lowest.

Overall prevalence of any abuse amounted to 23.2%. The highest prevalence rates of minor acts concerned psychological abuse (19.5%), followed by financial (2.6%) and physical abuse (2.5%), with the lowest being injury (0.4%) and sexual abuse (0.3%). The highest prevalence figures, for severe acts, concerned again psychological abuse (8.2%), followed by financial abuse (2.5%), and the lowest were physical abuse (1%), injury (0.3%) and sexual abuse (0.2%). Prevalence regarding any abuse was 21.3% for minor acts, and 10.8% for severe episodes.

## Chronicity/High chronicity of abuse

Regarding abuse chronicity, as mean value of frequency of acts during the past year (from 1–25) (Table 2), total ranged from 4.96 (financial) to 18.75 (psychological), whereas minor chronicity ranged from 3.37 (injury) to 15.56 (psychological), and severe chronicity from 1.60 (injury) to 9.44 (psychological). Chronicity figures, for total of any abuse, were 17.92, with minor at 15.43 and severe at 8.31.

Total high chronicity (% of cases above median) was more common concerning injury (62.5%), psychological abuse (47.8%), any abuse (44.7%), and financial abuse (18.5%), and less common concerning physical abuse (17%) and sexual abuse (16.7%).

## Factors associated with high chronicity of any abuse

As shown in Table 3, of the variables in the demographic/socio-economic block, participants from Germany, Greece, Italy, Portugal and Spain were at lower "risk" of high chronicity of any abuse. None of the variables in the life-style and social support blocks were independently associated with high chronicity of any abuse.

Of the variables in the health block, anxiety was significantly associated with a greater "risk" of high chronicity of any abuse, while of the perpetrator block, being a spouse/cohabitant and female were associated with higher odds. Overall, the model explained 34.4% of the variance in high chronicity of any abuse, of which 19.9% was related to demographic/socio-economic block, 0.6% life-style, 4.4% health, 0.3% social support, and 9.2% perpetrator, respectively.

## Discussion

This study aimed at describing the prevalence, chronicity and severity of abuse (i.e. psychological, physical, injury, sexual, financial, and any abuse), experienced by men aged 60–84 years in seven European countries, during the past 12 months, and at examining factors associated

**Table 2. Prevalence and chronicity of minor, severe and total abuse experienced by older men.**

| Abuse type | n (1908) | % Mean(SD) | Abuse type | n (1908) | % Mean(SD) |
|---|---|---|---|---|---|
| **Psychological** | | | **Sexual** | | |
| *Minor*[a] | 371 | | *Minor*[a] | 5 | |
| Prevalence | | 19.5 | Prevalence | | 0.3 |
| Chronicity | | 15.56(22.04) | Chronicity | | 4.40(5.94) |
| High chronicity[b] | | 45.8 | High chronicity[b] | | 80.0 |
| *Severe*[a] | 157 | | *Severe*[a] | 4 | |
| Prevalence | | 8.2 | Prevalence | | 0.2 |
| Chronicity | | 9.44(14.49) | Chronicity | | 2.50(1.00) |
| High chronicity[b] | | 38.9 | High chronicity[b] | | 25.0 |
| *Total*[c] | 387 | | *Total*[c] | 6 | |
| Prevalence | | 20.3 | Prevalence | | 0.3 |
| Chronicity | | 18.75(29.19) | Chronicity | | 5.33(6.80) |
| High chronicity[b] | | 47.8 | High chronicity[b] | | 16.7 |
| **Physical** | | | **Financial** | | |
| *Minor*[a] | 47 | | *Minor*[a] | 50 | |
| Prevalence | | 2.5 | Prevalence | | 2.6 |
| Chronicity | | 5.98(11.90) | Chronicity | | 4.84(11.54) |
| High chronicity[b] | | 38.3 | High chronicity[b] | | 44.0 |
| *Severe*[a] | 18 | | *Severe*[a] | 47 | |
| Prevalence | | 1.0 | Prevalence | | 2.5 |
| Chronicity | | 2.17(1.82) | Chronicity | | 3.40(7.77) |
| High chronicity[b] | | 33.3 | High chronicity[b] | | 36.2 |
| *Total*[c] | 53 | | *Total*[c] | 81 | |
| Prevalence | | 2.8 | Prevalence | | 4.2 |
| Chronicity | | 6.04(11.39) | Chronicity | | 4.96(13.25) |
| High chronicity[b] | | 17.0 | High chronicity[b] | | 18.5 |
| **Injury** | | | **Any abuse**[d] | | |
| *Minor*[a] | 8 | | *Minor*[a] | 406 | |
| Prevalence | | 0.4 | Prevalence | | 21.3 |
| Chronicity | | 3.37(5.13) | Chronicity | | 15.43(23.36) |
| High chronicity[b] | | 62.5 | High chronicity[b] | | 44.2 |
| *Severe*[a] | 5 | | *Severe*[a] | 201 | |
| Prevalence | | 0.3 | Prevalence | | 10.8 |
| Chronicity | | 1.60(0.55) | Chronicity | | 8.31(13.86) |
| High chronicity[b] | | 60.0 | High chronicity[b] | | 45.1 |
| *Total*[c] | 8 | | *Total*[c] | 443 | |
| Prevalence | | 0.4 | Prevalence | | 23.2 |
| Chronicity | | 4.37(5.60) | Chronicity | | 17.92(30.36) |
| High chronicity[b] | | 62.5 | High chronicity[b] | | 44.7 |

[a] = severity was dichotomized in minor and severe acts

[b] = above median

[c] = total chronicity does not necessarily correspond to the sum of minor/severe chronicity as respondents may have been exposed to both

[d] = Abuse was assessed with 52 items based on the Conflict Tactic Scales-2 (CTS2) and the UK survey of elder abuse/neglect; SD = Standard Deviation.

with total high chronicity of any abuse among these individuals, in order to contribute to reduce the lack of reliable information on the topic. It is to premise that, due to paucity of previous studies on male elder abuse, and consequently to the lack of sufficient information on

**Table 3. Factors associated with total high chronicity of any abuse among older men (n = 1719).**

| Independent Variables | OR | 95% CI |
|---|---|---|
| **Victims' demographic-socio-economic block** | | |
| *Country*[a] | | |
| Sweden[b] (Stockholm) | 1 | |
| Germany (Stuttgart) | .243*** | .104-.568 |
| Greece (Athens) | .149** | .045-.491 |
| Italy (Ancona) | .285** | .113-.721 |
| Lithuania (Kaunas) | .590 | .228–1.529 |
| Portugal (Porto) | .186**** | .073-.474 |
| Spain Granada) | .286* | .093-.883 |
| *Age* [a] | | |
| 60-64[b] | 1 | |
| 65–69 | 1.261 | .566–2.810 |
| 70–74 | .908 | .417–1.981 |
| 75–79 | .846 | .323–2.213 |
| 80–84 | 1.599 | .570–4.490 |
| *Foreign background*[a] | | |
| No[b] | 1 | |
| Yes | .806 | .309–2.106 |
| *Marital status*[a] | | |
| Married-cohabitant[b] | 1 | |
| Alone-single | 5.532 | .614–38.838 |
| Divorced-separated | 5.944 | .972–28.357 |
| Widower | 3.868 | .583–25.672 |
| *Education*[a] | | |
| Low[b,c] | 1 | |
| Middle[d] | 1.978 | .961–4.072 |
| High[e] | 2.482 | .974–6.324 |
| *Profession*[a] | | |
| Blue-collar[b] | 1 | |
| Low white-collar | .939 | .407–2.165 |
| Middle-high white-collar | .606 | .215–1.704 |
| *Still on work*[a] | | |
| No[b] | 1 | |
| Yes | .791 | .224–2.802 |
| *Financial support*[a] | | |
| Pension[b] | 1 | |
| Working | .939 | .407–2.165 |
| Other income[f] | .606 | .215–1.704 |
| *Financial strain*[a] | | |
| No[b] | 1 | |
| Yes | .806 | .309–2.106 |
| *Lives with*[a] | | |
| Spouse-cohabitant[b] | 1 | |
| Spouse-cohabitant-other[g] | 1.179 | .379–3.667 |
| Alone | .386 | .059–2.532 |
| Other[h] | .160 | .017–1.495 |
| *Persons at home*[a] | | |
| More than 3 persons[b] | 1 | |
| 1 to 2 persons | 1.209 | .657–2.222 |
| **R[2] Change** | **19.9** | |
| **Victims' life-style block** | | |

*(Continued)*

**Table 3.** (Continued)

| Independent Variables | OR | 95% CI |
|---|---|---|
| *Alcohol use*[a] | | |
| No[b] | 1 | |
| Yes | .554 | .282–1.089 |
| *Smoking*[a] | | |
| No[b] | 1 | |
| Yes | .791 | .377–1.661 |
| **R² Change** | **.6** | |
| **Victims' health block** | | |
| *Somatic symptoms*[a,m] | | |
| High[b,l] | 1 | |
| Low | .931 | .524–1.655 |
| *Anxiety symptoms*[i,n] | 1.166**** | 1.073–1.267 |
| *Depressive symptoms*[i,n] | .962 | .875–1.058 |
| **R² Change** | **4.4** | |
| **Victims' social support block** | | |
| *Social support*[a,o] | | |
| High[b,l] | 1 | |
| Low | .889 | .519–1.524 |
| **R² Change** | **.3** | |
| **Perpetrators' block** | | |
| *Sex*[a] | | |
| Men[b] | 1 | |
| Women | 2.150* | 1.149–4.022 |
| *Relationship*[a] | | |
| Other[b,p] | 1 | |
| Spouse-cohabitant | 2.840** | 1.434–5.627 |
| **R² Change** | **9.2** | |
| **Total R²,[q]** | **34.4** | |

[a] = categorical variables

[b] = comparison category

[c] = less than primary school-primary school-similar

[d] = secondary school-similar

[e] = university-similar

[f] = e.g. sick pension

[g] = e.g. children

[h] = e.g. children

[i] = continuous variable

[l] = above median

[m] = GBB: Giessen Complaint List

[n] = HADS: Hospital Anxiety and Depression Scale

[o] = MSPSS: Multidimensional Scale of Perceived Social Support

[p] = e.g. children/grandchildren, other relatives, friends, neighbors, caring staff

[q] = Nagelkerke $R^2$: approximation to descriptive goodness-of-fit statistics, to measure the fit of the proposed logistic model; OR = odds ratio; CI = confidence interval

*p < .05

**p < .01

*** p < .001

****p < .0001.

this issue, comparisons have been performed with "similar" findings, i.e. findings from surveys which were different from ours in method, sample construction, and measurement [e.g. 56]. For the same reason, in the discussion also explanations of results, which relate to elder abuse in general against older people, are proposed, thus not concerning only older men as victims. In this respect, it is worth to clarify that our results refer to abuse in general, and not specifically to abuse in the context of IPV, as already stated in the "Measure" section, although some studies on IPV have been proposed for the discussion of male abuse. Moreover, since the high prevalence in men could be influenced by the sensitivity of considering abuse from at least a single episode, and due to the differences regarding methods and instruments for measuring elder abuse, which emerged from previous literature on the topic, the comparison between men and women has been discussed but it should be considered with caution. The aim of our study is indeed mainly to highlight the existence of abuse also towards men, by suggesting a possible (although incomplete) portrait of male abuse, without providing specifically peculiarities of abused men with regard to female gender, but however with some support from findings on female abuse. It is furthermore to keep in consideration that, previous authors, indicated how both adult men and women are victims of mistreatment [57], and the related similar reasons generally pertain to vulnerable conditions in late life [33].

## Prevalence of abuse

Our findings indicated that the highest prevalence rates of total abuse regarded psychological abuse (20.3%), followed by financial abuse (4.2%), physical abuse (2.8%), injury (0.4%), and sexual abuse (0.3%). The total of any abuse amounted to 23.2%. Regarding psychological abuse, previous studies found that men had more often been abused than women, specifically mainly concerning psychological (20% *vs*. 18.9%) [33], and also that older men were more than women exposed to emotional (13% *vs*. 7%) and physical (8% *vs*. 1%) abuse by strangers [58]. A further study also found that men were more likely to experience emotional and financial abuse [59]. Reporting of sexual abuse by older men remains very low, if compared with the same findings concerning women. This could be generally related to a "lack of disclosure by victims" [60], given that men (including older men) avoid speaking out about experiences of violence; however, it might be also related to the fact that they do not experience sexual abuse to the same extent as women, e.g. across all stages of their life. In this respect, some authors suggested that, in the case of lifetime abuse, sexual violence "concerns women almost exclusively" [61]. Further authors highlighted that, although IPV during lifetime is experienced by both aging men and women, however the memory of the experienced abuse seems vague and marginal for male victims, whereas it remains an experience still alive for female victims [62, 63]. The lower incidence of sexual abuse towards men generally might also indicate a "failure to screen" such episodes [60].

On the whole, our prevalence figures are higher than those reported by studies with comparable male subjects [4, 7–10]. Discrepancies in results may be due to methodological divergences. For instance, some authors [9] considered an event of psychological abuse only if 10 or more incidents had occurred, whereas our study considered each single episode of abuse as an incident. In another study [4], the occurrence of verbal/psychological abuse was measured with only 1 item, whereas in the present paper 11 items were used. Thus, the rates of psychological abuse may be under-estimated in those studies. This consideration suggests the lack of a common operational definition of abuse and of instruments with good psychometric characteristics [7], which seem conversely crucial in order to detect abuse against older men, a phenomen that exists but still remains not well investigated, detected and reported [28, 30, 32]. Moreover cultural/geographical differences, for instance, concerning awareness, perception

and disclosure of abuse, might also explain divergences in prevalence rates of elder abuse across studies in different countries [33].

## Severity, chronicity/high chronicity of abuse

In our study, the highest figure for severe acts was for psychological abuse (8.2%), followed by financial abuse (2.5%), and the lowest for sexual abuse (0.2%). Minor severity of any abuse was experienced by 21.3% of the participants, and the most severe acts by 10.8%. These figures seem higher than that from the *Study of Abuse and Violence Against Older Women* in five European countries (i.e. Austria, Belgium, Finland, Lithuania, and Portugal), reporting 16% of intermediate severity, and 6.5% of the most severe level of different forms of abuse and violence [56]. Anyway, also in this respect, our methodology and operational definition of abuse could have had a role.

Our results also showed that chronicity figures, for the total of any abuse, were 17.92, and the prevalence of high chronicity (above median) concerned injury (62.5%) at highest, followed by psychological abuse (47.8%), and sexual abuse (16.7%) at lowest. This is overall in line with some previous research (not specifically regarding men) reporting that a considerable proportion of community-dwelling older women experience abuse repeatedly, with emotional abuse (45%) the highest, and the sexual abuse (23%) the lowest [41, 64]. Moreover, Burnes and colleagues [65] revealed a general mean severity score of 6.76 for emotional abuse, and 3.89 for physical abuse, in a sample of older people from the USA. These researchers did not find any differences between women and men in terms of severity of abuse. However, they measured the severity of abuse only based on the frequency of abusive acts, but not the intensity (as minor and severe acts).

## Factors associated with high chronicity of abuse

In general, in line with De Donder and colleagues [56], the current study found that chronicity of any abuse was generally related to country of residence. However, in particular our results showed that, being from any country other than Sweden, seems linked to a lower risk for older men of exposure to high chronicity of abuse, whereas, conversely, previous research cited above [56] revealed that, compared to Lithuania, being from Portugal was positively related to frequent occurrence of several forms of abuse towards older women, while being from Finland (a Nordic country as Sweden) was "protective". These findings could pertain to diversities across countries in social structures, cultural values as well as gender rules/patterns affecting conducts and relations in society [29, 33]. For example, there could be accepted cultural norms to obviate conflicts *via* violence [33, 66]. Awareness, appraisal and disclosure of violence might also be influenced by social and cultural structures of the context, as anticipated above with regard to different prevalence rates across studies on elder abuse. However, the report on *Gender Equality Index* [67] has shown that, compared to other countries, Sweden has the highest levels of awareness/perception of cases of domestic violence on social networks/environments, a factor which is positively related to disclosure of violence itself, i.e. talking about such experiences with other people/someone else, and/or reporting it to any institution.

Moreover, according to the literature, awareness and perception of elder abuse seem generally more widespread in Northern than Southern Europe. This could be due (among other factors) to the fact that, in the former countries, earlier studies on the topic began since the 1980s-1990s [68]. Moreover a better social landscape can be found in these societies [69], in addition to a greater ability to manage problems regarding ageing and elder abuse [66]. This ability is to be intended also in terms of more available policies and strategies against elder abuse (more frequently addressed to the general population, than specific for older men), with ministries

financing related initiatives and projects [70]. In countries of the "Nordic model' [71], welfare services are indeed mainly publicly financed, with a provision based on universal rights, and with municipalities greatly involved in their implementation. In particular, good practices of prevention and information interventions, as anti-abuse policies and programs for reporting and supporting victims of abuse, are in great part available, for instance, in the following countries [72]: Sweden, where in 2009 general guidelines, for protecting children and women (including older women), were provided; Germany, where a Charter of Rights for people, in need of long-term care and assistance, has been developed in the period 2003–2005; and other Northern European countries (i.e. Ireland, Norway, and the Netherlands), where policies for general prevention, and support to potential elder victims of violence, are implemented. As regards Sweden (as a Northern country), also the whole context described above (more research on and policies against elder abuse) could be reflected in a greater awareness of the existence of abuse, and in a positive/cultural attitude and willingness towards a "potentially protected" reporting. These circumstances may lead to a possible greater prevalence of the phenomenon. In Sweden, however, also changes in long-term care (e.g. reduction of traditional institutional elder care and increase of families "not ready" to assist) could be responsible for further cases of elder abuse [73]. Conversely, in Southern European countries such as Italy, elder abuse still remains a "social taboo", and older adults usually do not report episodes of violence, especially when they depend on relatives for care and support. This happens also due to the lack of an appropriate legal framework and policies on elder abuse at national level [68, 74].

It is also worth considering that in our study the research design, as methodological approach to interviews, might have influenced respondents' behaviors of reporting abuse and results, according to the country of residence. We realized indeed face-to-face interviews to older people and, in two cities, (i.e. Stockholm and Stuttgart) respondents, who did not want to be interviewed "in place", received a questionnaire for self-administration. According to the literature, self-administered methods are more likely to be accepted by respondents when compared to face-to-face screens, in particular when older people and/or sensitive issues (e.g. elder abuse) are involved [75, 76]. This phenomenon is indeed often under-reported, especially by men, who may feel more ashamed and humiliated when interviewers ask the questions [33], in particular if another person/relative is present during the face-to-face interview. Thus, self-administration could have limited the non-response rate and conversely it could have increased the number of potential victims reporting episodes of abuse. In our study this might be the case, at least, for Sweden. Moreover, according to previous publications based on ABUEL data, the highest prevalence rates of overall male elder abuse were found in Sweden and Germany (respectively, 37% and 30%) [33].

The results of our study also revealed that anxiety symptoms were positively linked with high chronicity of any abuse. This is, on the whole, in line with more general previous research on male elder abuse [28, 33], showing that a low functional/health status could increment the probability of victimization in later life. In particular, Kosberg [28] reported that older men, suffering from physical/mental ill-health, experience victimization. One explanation could be that anxiety symptoms result in higher levels of dependency and frailty, which make older men vulnerable to mistreatment. Furthermore, one could assume that mental ill-health of the older person might result in situational violence, in which the abuser was provoked to commit the abusive act [33]. However, it is also possible that behavioural disturbances themselves represent the reaction of the older victims to violence [77, 78]. Anxiety could, in particular, be the negative health consequence of high chronicity abuse, as several studies indicate that repeated abuse may be a risk factor for higher levels of emotional distress [64, 79].

Finally, in our study, to report/have a spouse/cohabitant and female individual as perpetrator increased the "risk" for high chronicity of any abuse. Previous studies showed that generally shared living accommodation might increase the risk for abusive episodes [80, 81], and, in particular, older victims are often cohabiting with their abusers [82]. Such findings also suggest how difficult might be to provide interventions for treating cases of elder abuse, especially when a close relationship exists between victim and perpetrator. Moreover, this relation might influence disclosure of abuse negatively. Further research findings indicate that the majority of abuse towards older men occurs in domestic settings, which represent an example of possible context/accommodation for cohabitation, and that abused older men, in these settings, seem unwilling to report abuse, because of the feeling of shame, in particular when the perpetrator is female [29]. Some authors [33] have also shown that abused older males were more likely living alone, or only with a spouse/partner. Furthermore, they have observed that older men living with a lower number of cohabitants were more likely to experience psychological and physical abuse. Other authors [65] have reported that living alone with the perpetrator increased the odds of a severe level of emotional abuse, physical abuse, and neglect. This is, in particular, an association which disappeared in the presence of a non-perpetrator cohabitant. The findings from a prevalence study in Israel [10] similarly showed that older people living with partners had a greater risk of abuse, including mental one. In summary, when there is a close relationship between victim and perpetrator, and the latter is a family member, the former might be oppressed/manipulated, due to a possible vulnerable condition of dependence [33]. This happens particularly when perpetrators are burdened family caregivers [60].

## Limitations

The present study has some limitations that need to be acknowledged [33]. Firstly, the data were collected only in seven European urban centres, and consequently the respondents may not be representative of those living in non-urban centres and in other countries. Thus, generalizability of the results cannot be guaranteed. Secondly, the data collection was based entirely on self-reports from the older participants, and were not objectively confirmed. Thus, the results should be interpreted with caution, as recall bias may have occurred. Thirdly, older persons with deficiencies (e.g. dementia) were excluded from the study (not able to complete the survey appropriately), this further impacting on the degree to which the results can be generalized. Fourthly, the few respondents referring certain types of abuse (e.g. sexual) could indicate a "systematic under-reporting of abuse" [33], calling for additional caution in interpreting and generalizing the results. This study had indeed a relatively low response rate (45.2%) across countries, which may have resulted in an under-reporting bias, but this seems also related to general population-based studies addressing sensitive issues such as abuse. Fifthly, the cross-sectional nature of the study precluded the "establishment of causal links" [33] between the variables and of temporality regarding the respective associations. Longitudinal studies are therefore warranted, in order to check the relations which emerged between elder abuse and other dimensions. There are also some factors, influencing the prevalence of the phenomenon, which were not investigated in this study, and which might however increase the possibility of abuse, such as functional status by means of ADLs and IADLs (Activity of Daily Living and Instrumental Activity of Daily Living), and the need of caregiving/help. Also aspects pertaining the perpetrators could have an impact, for instance: poor psychological health (e.g. dementia, depression); drug/substance or alcohol misuse (which are often linked to verbal and financial abuse); hostility, aggression and stress due to caregiver burnout; and abuser dependency on the victim for accommodation/housing, financial support, and/or other assistance [60, 66, 83]. The comparability between data collected with face-to-face questionnaires and self-

administered ones (195 in Sweden and 134 in Germany) has not been controlled, and although literature in this respect is mixed [e.g. 84–86], this represents another limitation to be considered. We further did not provide a multiple imputation of missing values in the multivariable analysis (subjects excluded = 189), because the overall abuse profile of subjects excluded was not statistically different from the included ones. However, the only one statistically significant difference in this respect, regarding the occurrence of severe physical violence (1% for the included vs. 2.9% for the excluded, p = .029), needs to be highlighted. Finally, the questionnaires were not tested psychometrically, in order to provide a cross-cultural "measurement equivalence" (ME), thus allowing valid comparison across different populations [87]. Therefore, the results should be interpreted with caution, since elder abuse is a sensitive issue also depending on country perceptions.

Apart from these limitations, our study may have provided the following: further insights concerning the issue of elder abuse, and new insights concerning the chronicity of abuse against men; "a workable definition of abuse and validated instruments to assess the phenomenon" [33], which could be used, among others, by researchers; and findings that, policy makers and caring/health staff, could use to develop efficient intervention and prevention strategies against the abuse of older men.

We have also to highlight that our data are about over 10 years old (from 2009). However they can be generally considered still current and valid because, to our knowledge, there are not new/additional cross-country surveys on elder abuse in Europe relying on such a large sample (about 4500 individuals), in particular with the inclusion of countries (e.g. Italy) that are still little investigated with regard to this topic. Also, in a recent systematic review and meta-analysis on elder abuse prevalence [11] ABUEL findings still seem the more recent available at least for some European countries. Moreover, the large collection of dimensions has allowed to explore, in particular, male elder abuse, that still seems under-recognized. However, in more recent years other external changes may have occurred, for instance a major provision of policies for reporting and supporting victims of abuse, and a greater awareness of the phenomenon, also due to the dissemination of the results from studies on this crucial public health issue. Therefore, a current follow up of the same ABUEL study, might probably reveal a greater prevalence rate of abuse than that recorded in 2009 and, maybe, especially for older men.

## Conclusions

Research findings usually indicate women at higher risk of experiencing elder abuse, and men at higher risk to be perpetrators. However, it is to be considered that such analyses should be socially and theoretically contextualised, in order to better understand/explore "why and how gender matters in elder abuse" [88]. Our study substantiates older men´s exposure to abuse, and also it may have provided new insights in relation to severity and chronicity of the phenomenon. Incorporation of severity into elder abuse research can cover in particular also clinical aspects of the question [89]. For men, the prevalence of psychological abuse emerged relatively high, and also some figures on exposure to severe (e.g. psychological) and repeated acts (e.g. injuries and psychological) were rather important. Older men, similar to older women, are indeed exposed to abuse, and generally this relates to the vulnerability of older adults [33]. However, social rules and expectations, which traditionally frame men as stoic, might lead them to deny experience of mistreatment [33]. It seems thus important to provide psychological/emotional support, and encourage male victims to speak out about their abuse and to consider acceptable to have help and support [90]. It was further observed that high chronicity, of overall/any abuse, was related to country of residence, anxiety symptoms, and having a spouse/cohabitant as perpetrator. These results also suggest that the consideration of

the perpetrators may be of great help in analyzing abuse experiences [91]. Results furthermore indicate that cultural differences among countries, health and living arrangements of older men, should be kept in consideration when providing programmes and interventions to prevent and treat elder abuse. In particular, cultural/regional discrepancies in incidence and/or reporting, suggest that additional research is needed, in order to appropriately address existing cultural ideologies which could impact on abuse perceptions [60]. Moreover, our findings may encourage research to further address the issue of severity and chronicity of abuse of older men, in order to provide new implications for public health practice and policy making. To prevent aggressive relationships among older adults could also require more public investment in formal support [92]. To identify victims of abuse, including older men, is thus a fundamental starting point, in addition to provide environment for appropriate and assisted reporting [93].

## Supporting information

**S1 Checklist. STROBE statement—checklist of items that should be included in reports of** *cross-sectional studies.*
(DOC)

## Acknowledgments

The authors wish to acknowledge and express their appreciation to the staff of the Executive Agency for Health and Consumers (EAHC, currently named CHAFEA). Moreover, the authors are especially grateful to the older people who participated in the study, for their kindness, efforts and answers.

**Disclaimer:** The authors have adapted some brief parts of the text from their own previous publication concerning the same main ABUEL Study, with appropriate attribution. The adapted text refers to some general socio-demographic information of the sample population, 'Materials and Methods' (as well established protocols), and 'Limitations'. The previous publication is the following: Melchiorre MG, Di Rosa M, Lamura G, Torres-Gonzales F, Lindert J, Stankunas M, et al. (2016) Abuse of Older Men in Seven European Countries: A Multilevel Approach in the Framework of an Ecological Model. PLoS ONE 11(1): e0146425. https://doi.org/10.1371/journal.pone.0146425.

## Author Contributions

**Conceptualization:** Maria Gabriella Melchiorre, Mirko Di Rosa, Joaquim J. F. Soares.

**Data curation:** Maria Gabriella Melchiorre, Mirko Di Rosa, Joaquim J. F. Soares.

**Formal analysis:** Mirko Di Rosa, Bahareh Eslami.

**Funding acquisition:** Francisco Torres-Gonzales, Mindaugas Stankunas, Jutta Lindert, Elisabeth Ioannidi-Kapolou, Henrique Barros, Giovanni Lamura, Joaquim J. F. Soares.

**Investigation:** Maria Gabriella Melchiorre, Gloria Macassa, Francisco Torres-Gonzales, Mindaugas Stankunas, Jutta Lindert, Elisabeth Ioannidi-Kapolou, Henrique Barros, Giovanni Lamura, Joaquim J. F. Soares.

**Methodology:** Maria Gabriella Melchiorre, Mirko Di Rosa, Gloria Macassa.

**Project administration:** Maria Gabriella Melchiorre, Francisco Torres-Gonzales, Mindaugas Stankunas, Jutta Lindert, Elisabeth Ioannidi-Kapolou, Henrique Barros, Giovanni Lamura, Joaquim J. F. Soares.

**Resources:** Maria Gabriella Melchiorre, Mirko Di Rosa, Bahareh Eslami.

**Software:** Mirko Di Rosa, Bahareh Eslami.

**Supervision:** Maria Gabriella Melchiorre, Gloria Macassa, Joaquim J. F. Soares.

**Validation:** Gloria Macassa, Francisco Torres-Gonzales, Mindaugas Stankunas.

**Visualization:** Maria Gabriella Melchiorre, Mirko Di Rosa, Joaquim J. F. Soares.

**Writing – original draft:** Maria Gabriella Melchiorre, Bahareh Eslami.

**Writing – review & editing:** Maria Gabriella Melchiorre, Mirko Di Rosa, Gloria Macassa, Bahareh Eslami, Francisco Torres-Gonzales, Mindaugas Stankunas, Jutta Lindert, Elisabeth Ioannidi-Kapolou, Henrique Barros, Giovanni Lamura, Joaquim J. F. Soares.

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
