## [Decision Letter · Decision Letter 0]

15 Feb 2021

PONE-D-20-37603

The Prevalence, Severity and Chronicity of Abuse towards Older Men: Insights from a Multinational European Survey

PLOS ONE

Dear Dr. Di Rosa,

Thank you for submitting your manuscript to PLOS ONE. After careful consideration, we feel that it has merit but does not fully meet PLOS ONE’s publication criteria as it currently stands. Therefore, we invite you to submit a revised version of the manuscript that addresses the points raised during the review process.

We look forward to receiving your revised manuscript.

Kind regards,

Chaisiri Angkurawaranon

Academic Editor

PLOS ONE

Journal Requirements:

4. We noticed you have some minor occurrence of overlapping text with the following previous publication(s), which needs to be addressed:

- https://doi.org/10.1371/journal.pone.0146425

In your revision ensure you cite all your sources (including your own works), and quote or rephrase any duplicated text outside the methods section. Further consideration is dependent on these concerns being addressed.

Reviewers' comments:

Reviewer's Responses to Questions

**Comments to the Author**

1. Is the manuscript technically sound, and do the data support the conclusions?

Reviewer #1: Yes

Reviewer #2: Yes

2. Has the statistical analysis been performed appropriately and rigorously? 

Reviewer #1: Yes

Reviewer #2: Yes

3. Have the authors made all data underlying the findings in their manuscript fully available?

Reviewer #1: Yes

Reviewer #2: Yes

4. Is the manuscript presented in an intelligible fashion and written in standard English?

Reviewer #1: Yes

Reviewer #2: Yes

5. Review Comments to the Author

Reviewer #1: Thank you for giving an opportunity to review this article. The study aims were to describe the prevalence, chronicity and severity of an abuse type experienced by older men in seven European countries, and to examine factors associated with high chronicity of any abuse type. The article is well written, and I also appreciate that authors addressed all those limitations of the study. The study is a bit old (conducted in 2009), but still providing interesting results. I would recommend it for publication within PLOS ONE which I have some minor comment and suggestions.

-Did authors assess participants overall functional status like ADLs/IADLs , participants’ underlying diseases, or the need of caregiver? They might increase risk of being abused.

-Was there any report of caregiving neglect? If so, what type of abuse was it categorized into?

-If there were more than one types of abuse occurred in one time, how would you count the event in terms of prevalence and frequency of acts? Count one for each type or count only the most severe type?

Reviewer #2: The authors present a paper analyzing the prevalence, chronicity and severity of different types of abuse in men over 60 years of age in seven European cities. Also, they describe the factors associated with the high chronicity of these types of abuse. Their results emphasize the importance of considering abuse of men in this age group in light of the factors associated with a high risk of chronicity. The manuscript was written well and the work is clearly and accurately presented. However, there are some comments to consider, hoping that they may improve some aspects of this study.

Introduction

1.Line 57-59: Authors indicates: “The overall prevalence of elder abuse varies between 0.6-55%, and this is due to various factors (e.g. socio-demographics characteristics) [7]. In particular, among the general population or community samples, elder abuse rates vary between 0.2-27.5% [7-10].” The wording can be a bit confusing, what does first prevalence and second prevalence refer to? Are the authors referring to different issues? I recommend further clarification of these data. Just as a suggestion, have the authors considered including prevalence data from the World Health Organization? (e.g., https://www.who.int/es/news-room/fact-sheets/detail/elder-abuse)

Method

2.Line 451-462: the authors indicate that there may be differences in administering surveys by face-to-face and self-administered interview. As the authors point out, the self-administered form might offer more privacy for reporting potential abusive situations, which could increase prevalence. This raises several questions that I would like to share: To what extent can these two methods of data collection influence the other variables assessed apart of reported abuse? Could the results collected through the two procedures be considered together? Have the authors thought of any way to control/check this?

3.Line 113: The authors indicate the following information: “including measures/tests if not already available/validated, were translated into the native languages, back-translated and culturally adapted.” What kind of cultural adaptation has been made and were these adaptations made for this study? This should be taken into account considering the cultural differences that may exist between the cities in which the evaluation has been carried out.

4.Line 223-228: I recommend authors to review whether it is appropriate to use ordinal alpha instead of Cronbach's Alpha to report on the internal consistency of the instruments they have applied (e.g., GBB, HADS, ...).

5.Line 148: For clarity, I recommend that the authors indicate the type of response for the Abuse dimension assessed with the CTS2. In case it was only (yes/no), please add this information.

6.Line 265: the authors indicate that missing values were excluded from the analyses. Did they consider imputing missing values to avoid elimination bias? What percentage of missing cases did they find in the self-administered questionnaires in Stockholm and Stuttgart? What was the criterion for considering valid responses (e.g., 75% of the HADS responses...)?

Results

7.In Table 1: it is indicated in the variable “Lives with”: Spouse/cohabitant and Spouse/cohabitant/other, what is the difference between "spouse/cohabitant" and "spouse/cohabitant/other"? Further explanation is needed.

Discussion

8.I recommend to the authors to clarify broadly that these results refer to abuse in general, not to abuse in the context of IPV. Besides, the high prevalence in men could be influenced by the sensitivity of considering abuse (from a single episode). Because of the methodological differences and the considered measures of abuse, the discussion and comparison between men and women should be conducted carefully. Please, clarify and revised those aspects further.

Minor;

-Line 102, I recommend to authors include the months of data collection.

-In Table 1, Psychological Chronicity is reported “18.75(29.19)”, but in physical cases “5.98±11.90”, please correct this typo error.

-Format needs a lot of work (i.e., periods, commas, italics, spaces, zeros…) and should be carefully revised (e.g., Wrong upper-case letter use after periods in line 553-554…).

6. PLOS authors have the option to publish the peer review history of their article (what does this mean?). If published, this will include your full peer review and any attached files.

Reviewer #1: No

Reviewer #2: No

---

## [Author Response · Author response to Decision Letter 0]

9 Mar 2021

Letter to Editor

PONE-D-20-37603

The Prevalence, Severity and Chronicity of Abuse towards Older Men: Insights from a Multinational European Survey

PLOS ONE

Dear Professor Chaisiri Angkurawaranon

PLoS ONE Academic Editor

We would like to thank the academic Editor and the Reviewer(s) for the insightful and valuable comments as to how our manuscript could be improved. We carefully changed the text as advised, according with the recommended "Major Revision". A marked-up copy of the changes made from the previous article file, as a 'Revised Manuscript with Track Changes' file is provided, in addition to an unmarked version without tracked changes, as “Manuscript”. Any changes have been also highlighted in the revised version of the manuscript. 

In this letter (below) the authors presented their responses to each point brought up by the Reviewer(s) are presented. Numbers of pages and lines (which are indicated in the “Revision/comments of the authors”) regard the version of the Manuscript with track changes.

We aim to submit this Manuscript for publication as a Research Article in PLoS ONE. 

Looking forward to hearing from You,

Your Sincerely,

Mirko Di Rosa, on behalf of the co-authors

Revision/comments of the authors: 

We checked again and adapted this accordingly, as follows: first line indent from 1,00 to 0,6 (apart from first sentence of each paragraph). 

Revision/comments of the authors: 

The two sections don’t match because “Funding Information” is right, whereas “Financial Disclosure” is wrong regarding the following sentence to be deleted.: “This study was partially supported by Ricerca Corrente funding from Italian Ministry of Health to IRCCS INRCA (GL)”. Thus we would kindly ask to make changes to our financial disclosure, and to include our right/updated statement as follows (we have also indicated this point in the cover letter): “The ABUEL Project, "Elder Abuse: A multinational prevalence survey." was supported by the European Commission, through the Executive Agency for Health and Consumers (EAHC, currently CHAFEA, Consumers, Health, Agriculture and Food Executive Agency, https://ec.europa.eu/chafea/index_en.htm), Public Health Programme 2008–2010 (Grant Agreement n. 2007123). The Project was awarded to JFS, FTG, MS, JL, EIK, HB, and GL. The funders had no role in study design, data collection and analysis, decision to publish, or preparation of the manuscript”.

Revision/comments of the authors: 

When we have submitted our dataset to Dryad, it has been assigned a unique (currently private) identifier, (doi:10.5061/dryad.f7m0cfxtt). We have however already provided this during the first submission of the manuscript to the journal, in our Data Availability Statement, as follows: “All relevant data are available from Dryad Dataset, https://doi.org/10.5061/dryad.f7m0cfxtt” . We have also provided (in a separate file) a temporary link for editors and reviewers: https://datadryad.org/stash/share/7TySKLIzgz_ZbC92pL4Cxk3liZMZ03JnuOwKNQFC1lU. Should our manuscript be accepted for publication, we’ll communicate this to Dryad in order to have the original DOI confirmed and available for the Journal. In other words, the DOI number will remain the same but it will be accessible. 

4. We noticed you have some minor occurrence of overlapping text with the following previous publication(s), which needs to be addressed: - https://doi.org/10.1371/journal.pone.0146425 In your revision ensure you cite all your sources (including your own works), and quote or rephrase any duplicated text outside the methods section. Further consideration is dependent on these concerns being addressed

Revision/comments of the authors: 

We have addressed occurrence of overlapping text with the previous publication (https://doi.org/10.1371/journal.pone.0146425). by citing the source (current ref. n. 33) and rephrasing when necessary any duplicated text outside the methods section. However, besides “Materials and Methods’ section, also some general socio-demographic information of the sample population, and some parts included in the ‘Limitations’, remain similar, as already stated in our “Disclaimer”.

Reviewers' comments:

Reviewer #1: 

Thank you for giving an opportunity to review this article. The study aims were to describe the prevalence, chronicity and severity of an abuse type experienced by older men in seven European countries, and to examine factors associated with high chronicity of any abuse type. The article is well written, and I also appreciate that authors addressed all those limitations of the study. The study is a bit old (conducted in 2009), but still providing interesting results. I would recommend it for publication within PLOS ONE which I have some minor comment and suggestions.

- Did authors assess participants overall functional status like ADLs/IADLs, participants’ underlying diseases, or the need of caregiver? They might increase risk of being abused.

Revision/comments of the authors: 

In our main study ABUEL we have not addressed functional status like ADLs/IADLs, but we have however mainly addressed aspects of health by means of other scales (e.g. HADS, GBB). Regarding the need of help, we used the MSPSS on perceived social support. This provided information on help from family, significant others, and friends, e.g. My family really tries to help me, I can count on my friends, and so on. We also collected information on diseases, care services and health care use, use of medication, but we have not analyzed all these dimensions for the manuscript. Reviewer #1 is right when highlights aspects (e.g. functional status by means of ADLs and IADLs, the need of caregiving/help might) which might increase risk of being abused. Thus we included a related sentence in the “Limitation section” of the paper (p. 30, lines 557-559). 

- Was there any report of caregiving neglect? If so, what type of abuse was it categorized into?

Revision/comments of the authors: 

In our main study ABUEL we addressed also neglect and we found a prevalence of 3% (0,6 for men, and 2,3 for women). Neglect was assessed with 13 items where the participants were asked whether they needed help/received it, needed help/did not receive it or did not need help with regard to the following aspects: Shopping, Preparing meals, Doing routine housework, Travel or transport, Getting in and out of bed, Washing or bathing (including getting in and out of bath or shower), Dressing or undressing Eating (including cutting up food), Getting to and using toilet, Help with correct dose and timing of medication, Any other day-to-day activity, Other household activities (e.g. gardening), General mobility in the house. However, regarding neglect we have not considered also chronicity and severity, and thus we have not included it in the manuscript. 

- If there were more than one types of abuse occurred in one time, how would you count the event in terms of prevalence and frequency of acts? Count one for each type or count only the most severe type?

Revision/comments of the authors: 

We counted one for each type/act. In this respect we had already provided a footnote in Table 2 in order to explain that total chronicity does not necessarily correspond to the sum of minor/severe chronicity as respondents may have been exposed to both. Anyway, probably this was not fully clear. Thus we have integrated this clarification in the main text of the manuscript (p.8, lines 183-184).

Reviewer #2: The authors present a paper analyzing the prevalence, chronicity and severity of different types of abuse in men over 60 years of age in seven European cities. Also, they describe the factors associated with the high chronicity of these types of abuse. Their results emphasize the importance of considering abuse of men in this age group in light of the factors associated with a high risk of chronicity. The manuscript was written well and the work is clearly and accurately presented. However, there are some comments to consider, hoping that they may improve some aspects of this study.

1. Introduction .Line 57-59: Authors indicates: “The overall prevalence of elder abuse varies between 0.6-55%, and this is due to various factors (e.g. socio-demographics characteristics) [7]. In particular, among the general population or community samples, elder abuse rates vary between 0.2-27.5% [7-10].” The wording can be a bit confusing, what does first prevalence and second prevalence refer to? Are the authors referring to different issues? I recommend further clarification of these data. Just as a suggestion, have the authors considered including prevalence data from the World Health Organization? (e.g., https://www.who.int/es/news-room/fact-sheets/detail/elder-abuse)

Revision/comments of the authors: 

In order to clarify this part of the Introduction, we have rephrased it and also we have included the reference suggested by Reviewer #2, i.e. Yon et al., 2017 (current ref. number 11), whose study was cited in the WHO link (p.3, lines 57-64). We have also updated the reference list accordingly.

2. Method Line 451-462: the authors indicate that there may be differences in administering surveys by face-to-face and self-administered interview. As the authors point out, the self-administered form might offer more privacy for reporting potential abusive situations, which could increase prevalence. This raises several questions that I would like to share: To what extent can these two methods of data collection influence the other variables assessed apart of reported abuse? Could the results collected through the two procedures be considered together? Have the authors thought of any way to control/check this?

Revision/comments of the authors: 

The main study ABUEL was an exploratory/pilot study of the prevalence on elder abuse, and it was one of the first to include some unexplored countries in this respect (e.g. Italy). Thus we were interested in reaching respondents as much as possible, by means of different ways of administration of the questionnaire. Regarding this general aim, as we already stated in the manuscript, the self-administered form (195 in Sweden and 134 in Germany) might offer more privacy for reporting potential abusive situations, which could increase reporting and prevalence. Anyway, interviewer-administered questionnaires could also allow high response rates (if compared to self-administered ones) when the presence of the interviewer may be of help in order to understand better some questions, whereas respondents alone could in some cases fill in the questionnaire leaving out some questions. In other words, these opposite effects could balance, allowing to integrate the outcomes from the two methodologies together. Literature in this respect seems various. Some studies indicate that self-administered questionnaires and face-to-face interviews provide similar estimates (e.g. Siemiatycki J, A comparison of mail, telephone, and home interview strategies for household health surveys. Am J Public Health 1979, 69: 238–245). Other studies stressed the importance of face-to-face interviews (e.g. O'Toole BI, Battistutta D, Long A, Crouch K: A comparison of costs and data quality of three health survey methods: mail, telephone and personal home interview. Am J Epidemiol 1986, 124: 317–328). Further authors compare the success of face‐to‐face, phone and web/self-administered data collection modes, on the topic of crime victimization, and suggest that a mixed‐mode strategy could be also recommended, since a good mixed‐mode strategy could lead to higher response rates and lower non‐response bias (Laaksonen S, Heiskanen M. Comparison of three survey modes, University of Helsinki Department of Social Research, Working Paper 2013;2). However, the conceptual issue regarding the comparability between data collected with interviewer- and self-administered questionnaires remains, as highlighted by Reviewer #2. Some authors in this respect suggest the following “In order to avoid any potential for bias in the comparison between the two modes, the sample was randomly divided into two equal groups and the comparison between the self-administered and the original interviewer-administered Child-OIDP was based on comparisons between the two groups…. In addition to their random selection, the comparability between the two groups was also established by showing no differences in their socio-demographic” (Tsakos, G., Bernabé, E., O'Brien, K. et al. Comparison of the self-administered and interviewer-administered modes of the child-OIDP. Health Qual Life Outcomes 6, 40 (2008). https://doi.org/10.1186/1477-7525-6-40). We haven’t provided such a control and we don’t know exactly to what extent these two methods of data collection influence the other variables assessed apart of reported abuse, although we could maybe suppose that, for instance, people with a higher level of education may be more willing and able to complete the questionnaire themselves, without the help of an interviewer. Thus we have addressed this question by adding a limitation to our manuscript (p.30, lines 563-566). We have also updated the reference list accordingly (current refs. 84-86).

3. Line 113: The authors indicate the following information: “including measures/tests if not already available/validated, were translated into the native languages, back-translated and culturally adapted.” What kind of cultural adaptation has been made and were these adaptations made for this study? This should be taken into account considering the cultural differences that may exist between the cities in which the evaluation has been carried out.

Revision/comments of the authors: 

In order to address this point, as requested by Reviewer #2, we have integrated the “Study design” section with more detailed information on the translation and cultural adaptation procedures (pp. 5-6, lines 125-139). However, since we did not provide the evaluation of the psychometric properties of questionnaires by the measurement equivalence (ME) between the different language versions of the tool, we included a further limitation of the manuscript, to highlight the lack of ME for valid comparison and interpretation of cross-cultural data/results. We also included accordingly a reference on ME (current ref. 87, Davidov E, Meuleman B, Cieciuch J, Schmidt P, Billiet J. Measurement equivalence in cross-national research. Annu Rev Sociol. 2014;40: 55-75. doi:10.1146/annurev-soc-071913-043137) (p. 30, lines 570-574). We have also updated the whole reference list accordingly.

4. Line 223-228: I recommend authors to review whether it is appropriate to use ordinal alpha instead of Cronbach's Alpha to report on the internal consistency of the instruments they have applied (e.g., GBB, HADS, ...).

Revision/comments of the authors: 

Following the suggestion by Reviewer #2, we calculated Ordinal Alpha for GBB and HADS, because it is a measure of internal consistency suitable for ordinal response scales with five or fewer options. Moreover, there is evidence that Cronbach’s Alpha tends to underestimate the value of these measure, and this is also our case. Thus we have updated the related values in the main text (p.11, lines 253-255).

5. Line 148: For clarity, I recommend that the authors indicate the type of response for the Abuse dimension assessed with the CTS2. In case it was only (yes/no), please add this information.

Revision/comments of the authors: 

For each type of event/abuse, the possible response was not only “Yes/No”. For each type we asked directly if it occurred once, twice, three to five, six to ten, 11-20 or .20 times during the past year, as explained in the section of Measures on “chronicity”. Each reported frequency was recorded as “Yes”. In order to clarify better the point, we have however integrated accordingly the main text in this respect (p. 8, lines 187-189). 

6. Line 265: the authors indicate that missing values were excluded from the analyses. 

- Did they consider imputing missing values to avoid elimination bias? 

Revision/comments of the authors: 

Subjects excluded from the multivariable analysis, due to missing data in the independent variables, were 189. We did not opt for multiple imputation of missing values because the overall abuse profile of subjects excluded was not statistically different from the included ones. The only statistically significant difference, between subjects included and excluded from the multivariable analysis, was about the occurrence of severe physical violence (1% for the included vs. 2.9% for the excluded, p=.029). This has been stated in the limitation paragraph (p. 30, lines 566-570). We have also corrected the n. indicated in Table 3 (1719 instead of 1908; line 357) and added the information on 189 subjects excluded in the “Data analyses” section (p. 12, line 294).

- What percentage of missing cases did they find in the self-administered questionnaires in Stockholm and Stuttgart? 

Revision/comments of the authors: 

As regards self-administered questionnaires, they were 195 in Sweden and 134 in Germany. Subject were excluded from the multivariable analysis because missing data in the independent variables were few, respectively 25 in Sweden and 29 in Germany of the self-administered ones. We have also added the information on these subjects in the “Data analyses” section (p. 12, line 294).

- What was the criterion for considering valid responses (e.g., 75% of the HADS responses...)?

Revision/comments of the authors: 

For the GBB there was no problem with missing items, since they were directly recoded into “not at all” category (possible answer to the question: “How much does each complaint discomforts you?”). Regarding MSPSS and HADS, the related data were used when 100% of the items were collected. We have also added this information in the “Data analyses” section (p. 12, lines 295-297).

7. Results In Table 1: it is indicated in the variable “Lives with”: Spouse/cohabitant and Spouse/cohabitant/other, what is the difference between "spouse/cohabitant" and "spouse/cohabitant/other"? Further explanation is needed.

Revision/comments of the authors: 

Regarding this point we had already included footnote “d” in Table 1, where we indicated that “other” could be for instance a daughter. Anyway it seems more clear to explain that “other” could include also “son, brother, sister, grandchildren”. Thus we have integrated the footnotes “d” and “e” (both regarding “other”) accordingly. 

8. Discussion I recommend to the authors to clarify broadly that these results refer to abuse in general, not to abuse in the context of IPV. Besides, the high prevalence in men could be influenced by the sensitivity of considering abuse (from a single episode). Because of the methodological differences and the considered measures of abuse, the discussion and comparison between men and women should be conducted carefully. Please, clarify and revised those aspects further.

Revision/comments of the authors: 

In order to address this point, in the premise to the whole Discussion we have addressed further the questions put in evidence by Reviewer #2, in order to clarify better that aim of the Discussion is to highlight the existence of abuse also towards men, without providing specifically peculiarities of abused men with regard to female gender, due to paucity and methodological differences of previous literature on the phenomenon. Thus, the comparison between men and women has been discussed but it should be considered with caution. (pp. 22-23, lines 381-391).

Minor:

- Line 102, I recommend to authors include the months of data collection.

Revision/comments of the authors: 

In this respect we have added January-July 2009 (p. 5, line 112).

- In Table 1, Psychological Chronicity is reported “18.75(29.19)”, but in physical cases “5.98±11.90”, please correct this typo error.

Revision/comments of the authors: 

We have deleted symbol “#” and we have included round brackets, as it is for the other values in the Table.

- Format needs a lot of work (i.e., periods, commas, italics, spaces, zeros…) and should be carefully revised (e.g., Wrong upper-case letter use after periods in line 553-554…).

Revision/comments of the authors: 

We thank very much Reviewer #2 for this further important comment. We have carefully revised the whole manuscript in this respect, in order to correct the format (in track changes).

---

## [Decision Letter · Decision Letter 1]

30 Mar 2021

The Prevalence, Severity and Chronicity of Abuse towards Older Men: Insights from a Multinational European Survey

PONE-D-20-37603R1

Dear Dr. Di Rosa,

We’re pleased to inform you that your manuscript has been judged scientifically suitable for publication and will be formally accepted for publication once it meets all outstanding technical requirements.

Kind regards,

Chaisiri Angkurawaranon

Academic Editor

PLOS ONE

Additional Editor Comments (optional):

Reviewers' comments:

Reviewer's Responses to Questions

**Comments to the Author**

1. If the authors have adequately addressed your comments raised in a previous round of review and you feel that this manuscript is now acceptable for publication, you may indicate that here to bypass the “Comments to the Author” section, enter your conflict of interest statement in the “Confidential to Editor” section, and submit your "Accept" recommendation.

Reviewer #1: All comments have been addressed

Reviewer #2: All comments have been addressed

2. Is the manuscript technically sound, and do the data support the conclusions?

Reviewer #1: Yes

Reviewer #2: Yes

3. Has the statistical analysis been performed appropriately and rigorously? 

Reviewer #1: Yes

Reviewer #2: Yes

4. Have the authors made all data underlying the findings in their manuscript fully available?

Reviewer #1: Yes

Reviewer #2: Yes

5. Is the manuscript presented in an intelligible fashion and written in standard English?

Reviewer #1: Yes

Reviewer #2: Yes

6. Review Comments to the Author

Reviewer #1: (No Response)

Reviewer #2: (No Response)

7. PLOS authors have the option to publish the peer review history of their article (what does this mean?). If published, this will include your full peer review and any attached files.

Reviewer #1: No

Reviewer #2: No

---

## [Editor Report · Acceptance letter]

5 Apr 2021

PONE-D-20-37603R1 

The Prevalence, Severity and Chronicity of Abuse towards Older Men: Insights from a Multinational European Survey 

Dear Dr. Di Rosa:

I'm pleased to inform you that your manuscript has been deemed suitable for publication in PLOS ONE. Congratulations! Your manuscript is now with our production department. 

Kind regards, 

on behalf of

Dr. Chaisiri Angkurawaranon 

Academic Editor

PLOS ONE